# Longitudinal CT Imaging to Explore the Predictive Power of 3D Radiomic Tumour Heterogeneity in Precise Imaging of Mantle Cell Lymphoma (MCL)

**DOI:** 10.3390/cancers14020393

**Published:** 2022-01-13

**Authors:** Catharina Silvia Lisson, Christoph Gerhard Lisson, Sherin Achilles, Marc Fabian Mezger, Daniel Wolf, Stefan Andreas Schmidt, Wolfgang M. Thaiss, Johannes Bloehdorn, Ambros J. Beer, Stephan Stilgenbauer, Meinrad Beer, Michael Götz

**Affiliations:** 1Department of Diagnostic and Interventional Radiology, University Hospital of Ulm, Albert-Einstein-Allee 23, 89081 Ulm, Germany; Christoph.lisson@uniklinik-ulm.de (C.G.L.); sherinachilles@aol.de (S.A.); marc.mezger@uni-ulm.de (M.F.M.); Daniel.wolf@uniklinik-ulm.de (D.W.); stefan.schmidt@uniklinik-ulm.de (S.A.S.); wolfgang.thaiss@uniklinik-ulm.de (W.M.T.); meinrad.beer@uniklinik-ulm.de (M.B.); michael.goetz@uni-ulm.de (M.G.); 2Center for Personalized Medicine (ZPM), University Hospital of Ulm, Albert-Einstein-Allee 23, 89081 Ulm, Germany; Ambros.Beer@uniklinik-ulm.de; 3Artificial Intelligence in Experimental Radiology (XAIRAD), Department of Diagnostic and Interventional Radiology, University Hospital of Ulm, Albert-Einstein-Allee 23, 89081 Ulm, Germany; 4Visual Computing Group, Institute of Media Informatics, Ulm University, James-Franck-Ring, 89081 Ulm, Germany; 5Department of Nuclear Medicine, University Hospital of Ulm, Albert-Einstein-Allee 23, 89081 Ulm, Germany; 6Department of Internal Medicine III, University Hospital of Ulm, Albert-Einstein-Allee 23, 89081 Ulm, Germany; johannes.bloehdorn@uniklinik-ulm.de (J.B.); Stephan.Stilgenbauer@uniklinik-ulm.de (S.S.); 7Center for Translational Imaging “From Molecule to Man” (MoMan), Department of Internal Medicine II, University Hospital of Ulm, Albert-Einstein-Allee 23, 89081 Ulm, Germany; 8i2SouI—Innovative Imaging in Surgical Oncology Ulm, University Hospital of Ulm, Albert-Einstein-Allee 23, 89081 Ulm, Germany; 9Comprehensive Cancer Center Ulm (CCCU), University Hospital of Ulm, Albert-Einstein-Allee 23, 89081 Ulm, Germany; 10German Cancer Research Center (DKFZ), Division Medical Image Computing, 69120 Heidelberg, Germany

**Keywords:** mantle cell lymphoma, tumour heterogeneity, imaging based texture analysis, risk assessment, prediction, relapse, personalized medicine, precision imaging

## Abstract

**Simple Summary:**

Mantle cell lymphoma (MCL) is considered an aggressive lymphoid tumour with a poor prognosis. However, according to recent studies, MCL is more heterogeneous than initially assumed with indolent subtypes without the need for immediate intervention. Currently, there are no routine biomarkers for the early prediction of relapse. The urge for personalized medicine has given rise to “radiomics”—the quantification of heterogeneity by imaging based texture analysis which has shown excellent results in numerous fields of application. Our study investigated the potential of CT-derived 3D radiomics as a non-invasive biomarker to risk-stratify MCL patients, thus promoting precision imaging in clinical oncology.

**Abstract:**

The study’s primary aim is to evaluate the predictive performance of CT-derived 3D radiomics for MCL risk stratification. The secondary objective is to search for radiomic features associated with sustained remission. Included were 70 patients: 31 MCL patients and 39 control subjects with normal axillary lymph nodes followed over five years. Radiomic analysis of all targets (*n* = 745) was performed and features selected using the Mann Whitney U test; the discriminative power of identifying “high-risk MCL” was evaluated by receiver operating characteristics (ROC). The four radiomic features, “Uniformity”, “Entropy”, “Skewness” and “Difference Entropy” showed predictive significance for relapse (*p* < 0.05)—in contrast to the routine size measurements, which showed no relevant difference. The best prognostication for relapse achieved the feature “Uniformity” (AUC-ROC-curve 0.87; optimal cut-off ≤0.0159 to predict relapse with 87% sensitivity, 65% specificity, 69% accuracy). Several radiomic features, including the parameter “Short Axis,” were associated with sustained remission. CT-derived 3D radiomics improves the predictive estimation of MCL patients; in combination with the ability to identify potential radiomic features that are characteristic for sustained remission, it may assist physicians in the clinical management of MCL.

## 1. Introduction

Mantle cell lymphoma (MCL) is a rare mature subtype of B cell non-Hodgkin lymphoma associated with an aggressive, or less frequently, indolent course. Despite the increasingly better understanding of the pathology and therapeutic options, the prognosis of MCL is generally poor, with 5-year survival rates of approximately 60% of the young, transplant eligible patients [1,2,3,4,5]. 

In general, the clinical outcome of the disease is highly heterogeneous [5,6]. Novel therapeutic options are constantly evolving, such as the CAR T-cell therapy Tecartus (brexucabtagene autoleuce, Santa Monica, CA, USA) [7]. However, until today MCL remains incurable. In general, clinical presentation and outcome of MCL are very heterogeneous with minimal symptoms or progressive generalized lymphadenopathy, splenomegaly, extranodal disease, and cytopenia, and thus require different treatment strategies [8,9,10,11]. 

As a consequence, the World Health Organization (WHO) updated the classification of MCL in 2017, describing two main subtypes, “classical MCL” and “indolent leukemic nonnodal/smoldering MCL,” with significant differences in the molecular characteristics, clinical features, prognosis and treatment options [12].

In 2008, the MCL International Prognostic Index (MIPI) was developed - the first MCL-specific prognostic model that incorporates age, performance status, lactate dehydrogenase (LDH), and leukocyte count [13]. 

The combination of the Ki-67 index and MIPI led to the more refined combined MIPI-c with a refined risk stratification, reflecting their strong complementary prognostic effects while integrating the most relevant prognostic factors available in clinical routine [4].

The promising approach to detect circulating tumour DNA (ctDNA) seems a highly promising biomarker in lymphoma with multiple landmark studies in aggressive B-cell lymphomas proving its potential as the most advanced prognostic factor nowadays [14].

In 2018, Lakhotia et al. reported the results of ct-DNA monitoring during DA_EPOCH-R chemotherapy in MCL, showing both a good correlation between the baseline ct-DNA level and the total tumour volume and the high ability of ct-DNA clearance after one cycle to predict a better PFS in those clearing rapidly the ct-DNA. However, due to the need for patient-specific primers and standardization issues measuring of ct-DNA level has not yet become established in the clinical practice of MCL yet [15]. 

Until now, no routine biomarkers have been established for early and accurate prognosis prediction [13,16]. 

According to the ESMO guidelines, there is consensus to perform a contrast-enhanced computed tomography (CT) scan of the neck, thorax, abdomen, and pelvis as initial staging [17]. A PET/CT scan is especially recommended in the rare limited stages I/II before localized radiotherapy, although reimbursement by domestic health insurance is not ensured. 

The emergence of technological innovation and the urge to fulfil personalized medicine has given rise to the constantly evolving field of research called “radiomics”—a computer-assisted technique for extracting and quantifying patterns, so-called radiomic features within diagnostic medical images to reflect the radiographic phenotype using data characterization algorithms. By capturing signal intensity distribution, i.e., grey-level patterns, radiomics quantifies a large panel of phenotypic characteristics, such as shape and texture, potentially reflecting intra- and intertumour heterogeneities [18,19,20,21,22,23]. 

Studies show that tumour heterogeneity is a central feature of malignancy as it contributes to treatment response, relapse, and overall survival [24,25,26,27,28,29]. Radiomics has shown promising results to analyze tumour heterogeneity using different imaging techniques, including artificial intelligence-based machine-learning algorithms [22,26,30,31,32,33,34,35,36]. Taking into consideration the “whole tumour volume” assessment across the entire body instead of just a tiny sample from a single site, radiomic analysis might represent a non-invasive prediction approach to identify patients at high risk of relapse; early identification of these patients could allow modification of their therapeutic management to reduce unnecessary toxicity and improve prognosis according to follow-up studies [18,20,37,38]. There have been other studies on the diagnostic value of radiomics regarding different types of lymphoma and the reproducibility of CT texture parameters [31,33,35,39,40]. 

However, to our knowledge, this is the first study to investigate the potential value of longitudinal CT-derived 3D texture analysis for early predictive estimation of MCL relapse based on radiomic changes during and after therapy. We, therefore, wanted to determine (1) whether CT-derived 3D radiomic features are predictive for relapse and (2) whether there are therapy-related changes in radiomic parameters that are characteristic for sustained remission without significant differences to normal lymph nodes.

## 2. Materials and Methods

### 2.1. Patients and Imaging Protocol

In this study, 31 treatment-naive patients with proven mantle cell lymphoma (by pathological reference assessment) who underwent contrast-enhanced CT or PET/CT scan at our institution before therapy initiation and with available clinical and in-house imaging follow-up data five years after the end of the first-line therapy were included in this retrospective study at our institution between January 2005 and December 2018. 

Demographic patient data, laboratory and clinical data (such as white blood cell count, lactate dehydrogenase levels, and Ki-67 proliferation), disease stage according to the Ann Arbor staging system, the existence of bulky disease (defined as a mass ≥ 10 cm in maximal diameter), treatment regimen and clinical outcome data were recorded by thoroughly reviewing electronic charts and the radiology information system. 

Incomplete clinical and imaging data records and lack of histological confirmation were exclusion criteria. 

Additional inclusion criteria were treatment with an R-CHOP-based regimen (rituximab, cyclophosphamide, doxorubicin, vincristine, and prednisone) alternating with or instead of an R-DHAP based regimen (rituximab, dexamethasone, high-dose cytarabine, and cisplatin) or R-FC (rituximab, fludarabine, and cyclophosphamide). Further exclusion criteria included patients whose disease status was not confirmed at the end of therapy and those who had disease progression within the first line of treatment.

All enrolled MCL subjects were followed up for at least five years, except those who experienced death of any cause. A flow diagram of the cohort selection is presented in Figure 1.

Patients in clinical remission and without disease residuals on contrast-enhanced CT or PET/CT scans were considered to be in complete response (CR). Imaging for assessment of disease status was performed close to therapy initiation (max. 3 weeks) and 3, 6, and 12 months after having therapy as part of the routine staging. 

As control cases, 39 treatment-naïve patients with non-small-cell lung carcinoma (NSCLC) and normal axillary lymph nodes confirmed by at least two contrast-enhanced PET/CT scans were identified from the radiology database from 2005 to 2010. The medical records of the control cases were reviewed in 2018 to ensure that these subjects honestly had normal axillary lymph nodes and did not harbor any hematological disease or axillary lymph nodes metastases that clinically manifested only several years later. 

Imaging of MCL patients before therapy initiation and their follow-up CT scans 3, 6, and 12 months after having started therapy and CT imaging of control subjects with normal axillary lymph nodes were analyzed. A total number of 745 target lymph nodes were eligible for inclusion. 

CT scans were performed in the cranial-caudal direction on a multiple-row detector CT scanner (Philips Brilliance CT 64-channel scanner and Philips Brilliance iCT 256-channel scanner, Philips Healthcare, Cleveland, OH, USA) with the following acquisition and reconstruction parameters according to standard protocol: tube voltage 100 kV–120 kV with an automatically calculated tube current; collimation 40 × 0.625 mm and 64 × 0.625 mm; matrix 512 × 512, 1 mm reconstruction thickness, and increment 0.5 mm after intravenous administration of (Ultravist® 370, Bayer Schering Pharma, Berlin, Germany) in weight-adopted dose with a delay of 70 s to represent the portovenous phase of chest and abdomen. Thin-slice CT scans were reconstructed using a soft tissue kernel (filter, B31f) for visual assessment and texture analysis. 

18F-FDG-PET/CT image acquisition was performed approximately 60 min after intravenous administration of 282 MBq of 18F-FDG (range 220-334 MBq). Patients fasted for at least six hours before the injection; PET/CT scans were carried out with a 40-slice CT with two overlapping X-ray beams and a 21.8 cm axial field of view PET detector Biograph mCT (40)S (Siemens Biograph mCT(40)S, Siemens Healthineers, Erlangen, Germany). The contrast-enhanced spiral CT scan was performed in the portal venous phase 80 s after intravenous contrast agent injection (Ultravist 370, Bayer Schering Pharma, Berlin, Germany) in a weight-adopted dose using attenuation-based online modulation of tube current (CARE Dose) with quality reference tube current setting (reference mAs) of 210 mAs, 120 kV, 16 × 1.2-mm collimation, a 512 × 512 matrix, and a 4 mm slice thickness followed by the PET scan from the mid-thighs to the base of the skull in 5 to 8 bed positions. All PET scans were acquired in 3D mode with an acquisition time of 3 min per bed position in time of flight technique. PET data were reconstructed with attenuation correction using dedicated standard software (PETsyngo, Siemens, Erlangen, Germany).

### 2.2. Image Processing and Feature Extraction

Images were evaluated by two experienced board-certified radiologists, with 10-year experience in oncologic imaging and over 5-year experience in segmentation and texture analysis. In cases of disagreement, consent was established by joint consultation. Target lesions were selected based on Cheson criteria [41]. Segmentation and texture analysis were performed using the software mint Lesion™ (mint Medical GmbH, Heidelberg, Germany), which allows three-dimensional size and whole lesion radiomic measurements at multiple times to examine temporal tumour heterogeneity. 

All 745 included target lesions were segmented in a semi-automatic process. Tumour boundaries were defined and manually contoured on axial CT images on the superior and inferior parts of the lesion. The software algorithm computed the contour on the slices in between for “whole tumour volume” data. 

Final 3D segmentation was thoroughly reviewed, and if necessary, the contour was manually modified (see Figure 2 for a depiction of the workflow). 

### 2.3. Statistical Analysis

Extracted radiomic features were tested as potential predictive factors. Lesion size measurements in one- and two dimensions and 3D whole lesion measurements were used as competing features. Textural features were selected based on the literature review and Spearman’s rank correlation coefficient test results [31,32,33,34].

20 out of 72 texture features of first- and second-order derived from the grey-level co-occurrence matrix were chosen for further analysis (see Table A2 in Appendix A). The selected parameters were limited to 10 features of first- and 10 of second-order to avoid overfitting [35]. First, the performance of the radiomic features was examined by using box-and-whisker plots as a graphical representation to assess the distribution of the extracted data set between CR (complete remission) and RD (relapse of disease). 

Second, the Mann-Whitney U test for non-normally distributed features was used to compare each selected texture feature across the two outcome groups.

A significance level of 0.05 was used for all statistical tests. Benjamini-Hochberg correction was used in each study to estimate true type-I error probability and to account for multiple hypothesis testing [42]. We used a spreadsheet that implements the Benjamini-Hochberg method for calculating the corrected significance level according to Weinkauf (https://github.com/WeinkMFG/MSExcel, accessed on 7 October 2021) as described by Chalkidou et al. [43]. The adjusted level of significance was 0.01 assuming a false discovery rate (FDR) of 0.05. 

Third, receiver operating characteristic (ROC) curve analysis was used to measure the performance of texture features to discriminate between the CR and the RD group for estimation of prognosis. The lower and upper limits of a 95% confidence interval were computed from generalized estimating equation. The area under the curve (AUC) was calculated for each feature. Optimal cutoff values were established with maximized sensitivity, specificity, and accuracy (percentage of correctly classified images/study objects as verified histologically) based on the Youden index to measure the effective clinical benefit in diagnostic decision-making.

Radiomic analysis of the target lesions was performed at several time points (pretherapy and follow-up imaging) to assess the modulation of the mutational profile over time. Statistically significant features for future relapse were regarded as potential predictive markers representing “MCL at high risk of relapse”. 

To identify potential radiomic features that are characteristic for sustained remission without a significant difference to normal lymph nodes, subgroup analyses of MCL patients with complete remission (CR) and MCL patients with relapse of disease (RD) versus a reference group (REF) with normal lymph nodes were performed by Mann-Whitney-U-test. The result was considered significant if *p*-value was <0.01 after applying the Benjamini-Hochberg correction for multiple testing. 

All statistical analyses were performed using IBM SPSS Statistics for Windows, Version 22.0 (IBM Corp., Armonk, NY, USA) [44]. 

## 3. Results

### 3.1. Patient Characteristics

A total of 70 consecutive patients, 31 patients with biopsy-proven MCL (5 women and 26 men; mean age 61.5 ± 9.7 years, range 42–76) and 39 patients with 18F-FDG-PET/CT-confirmed normal axillary lymph nodes as a control group (18 women and 21 men; mean age 64.9 years ± 8.5 range: 47–81) met the criteria for participation in the study.

Among the 31 MCL patients, 95% had a follow-up CT at 3 months, 76% at 6 months, and 49% at 12 months after having started (immune) chemotherapy +/− 4 weeks. 

2 patients (6.5%) of the CR group and 2 patients of the RD group died within the 5-year observation period (13% of the MCL patients). 

In the reference group, an average of 3.7 lymph nodes per patient (range 3–4, were analyzed and an average of 6.4 lymph nodes per patient (range 2–12) in the MCL group. In total, 745 target lesions were evaluated. In this cohort, 22 patients (71%) were responders and in complete remission. In comparison, 9 patients (29%) responded to first-line treatment but relapsed within the 5 year observation period. 

Nine patients received radiotherapy (29%). 

All patients’ baseline clinical characteristics, histologic subtype, distribution of disease stage at diagnosis, and treatment received are summarized in Table 1.

### 3.2. Radiomic Fingerprint of MCL at High Risk of Relapse

Of all features extracted using the designated method, measurements of the central tendency and dispersion of the radiomic parameter values showed no detectable differences between the relapse of disease (RD) and the complete remission (CR) group in the pretherapy and follow-up imaging at 3 months. 

After 6 months, however, the four texture features “Uniformity”, “Entropy”, “Skewness” and “Difference Entropy” showed predictive significance for relapse between the CR group and the RD group (two-sided Mann Whitney U test, *p* < 0.05, adjusted with Benjamini-Hochberg).

When these four texture parameters were further evaluated by conducting ROC analysis, all of them showed area under the curve (AUC) values with statistically significant differences (defined by *p* < 0.05). The greatest AUC values showed “Uniformity” (AUC 0.778) and “Entropy” (AUC 0.777). The detailed data are summarized in Table 2. 

To assess the clinical benefit, patients were classified above and below the cut-off values into low- and high-risk groups for relapse. Accuracy (percentage of correctly classified images as verified histologically), sensitivity, and specificity of the four statistically significant texture features above were calculated. 

“Uniformity” and “Entropy” had the highest discriminatory power to predict relapse with an optimal cut-off ≤0.0159 for “Uniformity” and a cut-off ≥6.2920 for “Entropy”, calculated by Youden’s index with 87% sensitivity, 65% specificity, 69% accuracy and a likelihood-ratio of 2.5 for “Uniformity” and with 80% sensitivity, 72% specificity, 73% accuracy and a likelihood-ratio of 2.9 for “Entropy”. 

Table 3 highlights all texture features that are strongly correlated to future relapse in MCL patients. It should be noted that for the texture features “Entropy”, “Skewness” and “Difference Entropy”, the higher the values, the higher the probability for future relapse. “Uniformity” behaves exactly the opposite way: lower values represent a higher probability of future relapse.

The ROC curve of the radiomic feature “Uniformity” shows good discrimination between the “complete remission” and the “relapse of disease” group in contrast to “Volume” without statistically significant discrimination (see Figure 3).

### 3.3. Radiomic Changes of MCL Lymph Nodes Characteristic for Sustained Remission

To identify potential radiomic features that are characteristic for sustained remission without significant differences to normal lymph nodes, analyses of MCL patients versus a reference group with normal axillary lymph nodes were performed by Mann-Whitney-U-test. The result was considered significant if *p*-value was <0.01 after applying the Benjamini-Hochberg correction for multiple testing. 

Subgroup analyses between the complete remission and the reference group (CR vs. REF) and between the relapse of disease and the reference group (RD vs. REF) were carried out to search for changes in radiomic features during therapy related to normal lymph nodes and sustained remission, respectively. 

For comparability and to minimize disruptive factors relating to topographic sites, only axillary lymph nodes were selected in both MCL groups and the reference group (MCL group with 24 lesions, reference group with 146 lesions). The reference group consisted of more patients in contrast to the MCL group as the number of lymph nodes from each patient eligible for analysis was much lower and to depict the nature of a normal lymph node as reliably as possible. 

Four different periods were analyzed: pretherapy (CR_pre_, RD_pre,_ and REF), 3 months follow-up (CR_1_, RD_1_, and REF), 6 months follow-up (CR_2_, RD_2_, and REF), and 12 months follow-up (CR_3_, RD_3_, and REF) imaging. 

All radiomic parameters, including lesion size measurements, showed significant differences in the pretherapy (CR_pre_ vs. REF and RD_pre_ vs. REF) and the follow-up imaging at 3 months (CR_1_ vs. REF and RD_1_ vs. REF). 

A split of both subgroup analyses occurred at the follow-up imaging at 6 months (CR_2_ vs. REF and RD_2_ vs. REF): In the analysis between the relapse of disease and the reference group (RD_2_ and REF), all radiomic parameters continued to differ significantly, whereas between the complete remission and the reference group (CR_2_ vs. REF) the following parameters showed no longer statistically significant differences suggesting normal lymph nodes in both groups:Sum averageAutocorrelationJoint averageShort axisVolumeP90thSkewness

This was confirmed at the subsequent follow-up imaging at 12 months with continued differences in the relapse vs. control (RD_3_ and REF) group in contrast to the remission vs. reference (CR_2_ vs. REF) group analysis (see Figure 4 and Figure 5 for examples). 

## 4. Discussion

This study employed a 3D radiomic analysis approach including first- and second-order texture features on pretreatment and follow-up routine CT imaging of MCL patients to investigate radiomic texture changes reflecting temporal tumour heterogeneity. Heterogeneity is associated with adverse tumour biology and correlates with molecular subtypes and clinical outcomes [25,27,45]. Studies prove the quantification of tumour heterogeneity as a promising tool for monitoring treatment response and clinical outcome [21,28,46]. Radiomics is an innovative field of computer-based research that reveals disease characteristics from medical images that are not visually seen to non-invasively quantify tumour heterogeneity for precision medicine [18,19,20]. Studies have shown that radiomics may improve the accuracy of diagnosis and prediction to support clinical decision-making using different imaging techniques in various malignancies, however, there is still limited evidence in Lymphoma, especially MCL [30,33,34,35,47,48,49,50,51,52,53]. 

Due to their complex intra- and intertumoural heterogeneity, diagnosis and choice of treatment are challenging in lymphoma [26,54]. For this reason, it is essential to analyze the heterogeneity within one type of lymphoma to identify potential image-based biomarkers for personalized cancer medicine. A few studies explored the potential value of radiomics as a diagnostic and prognostic tool in lymphoma with controversial results. Recently, several evaluation criteria and guidelines have been proposed to aid the assessment of radiomics research [20,55]. 

According to the ESMO guidelines, there is a consensus to perform a computed tomography (CT) scan of the neck, thorax, abdomen, and pelvis as initial staging [17]. Having this in mind, we chose CT-based texture analysis as it is easily accessible as a routine examination method. By measuring and analyzing target lesions in three dimensions, we identified several radiomic features to stratify the patient population in low- and high-risk relapse groups after 12 months with “Uniformity” and “Entropy” having the highest power to discriminate between MCL at low and high risk of relapse. Interestingly, the conventional lesion size measurements “Short Axis” and “Long Axis” as well as “Volume” could not discriminate between the future complete remission and the future relapse of disease group and were therefore not able to predict patient outcome. 

“Entropy” is a radiomic parameter that describes the degree of disorder regarding the distribution of the grey-level values. The study by Moon et al. demonstrated that “Entropy” using 18F-FDG-PET/CT based texture analysis correlated with genetic heterogeneity and mutation burden in lung cancer [56]. Choi et al. used dual-energy CT based radiomics, including entropy, which strongly correlated with the pathological heterogeneity index in lung cancer [57]. “Entropy” has been used for prognostic prediction in patients with high-risk oropharynx carcinoma after chemoradiation, in lung cancer patients after EGFR tyrosine kinase inhibitor treatment, and has been linked to treatment failure in metastatic colorectal cancer [36,58,59].

Regarding lymphoma, “Entropy” has been evaluated for outcome prediction in pediatric Hodgkin lymphoma using pretherapy 18F-FDG-PET/CT, and for predicting disease-free survival and overall survival in aggressive non-Hodgkin’s lymphoma without evidence being a statistically significant marker in either study [35,39]. However, the recent study by Mayerhoefer et al. demonstrated the potential value of “Entropy” on 18F-FDG-PET/CT scans for outcome prediction of MCL patients. This also includes the fact that a higher cut-off value of entropy represents a higher probability for future relapse in both studies, which supports the potential and reliability of this biomarker.

However, our study is the first to show that the texture feature “Uniformity” may be well suited for prognosis estimation in patients with MCL; MCL may be a “genomically unstable” disease with (sub)clonal heterogeneity regarding different topographic sites within an individual and with different modulation of the mutational profile over time [35,60]. No biomarkers are currently established for the prediction of clinical outcomes less than five years [13,16]. 

To the best of our knowledge, our study is the first to evaluate temporal heterogeneity in MCL patients for risk stratification earlier to the MIPI score used in clinical routine, which originated from data of 5-year survival. Our results could lead to biomarkers that enable clinicians to minimize (over)treatment or initiate a closer follow-up. Being able to assess the course of the disease at multiple sites non-invasively and time points might help to minimize the number of invasive procedures for histological analysis, the associated complications, and potential sampling errors due to heterogeneity. 

To understand MCL biology and changes of the MCL profile during therapy, high and low risk MCL groups were compared with normal lymph nodes from the same topographic site to maximize comparability. 

While all radiomic features of the analysis “MCL relapse versus normal lymph nodes” continued to significantly differ at all follow-up time points, between the complete remission and the reference group at the 6 month follow-up imaging, however, the parameters “Short Axis” and “Volume”, the first-order texture features “P90th” and “Skewness” and the second-order features “Autocorrelation”, “Joint Average” and “Sum Average” showed no longer statistically significant differences. This was confirmed at the follow-up imaging at 12 months with continued differences in the relapse vs. control group compared to the remission vs. reference group analysis. Further prospective studies with a larger cohort size are necessary. Still, our explorative findings might identify MCL patients with a lower risk of relapse where aggressive therapy and the resulting side-effects (e.g., secondary malignoma) could be saved.

Certainly, there are limitations to this study. The most apparent limitations were the retrospective study design and the moderate cohort size of patients which is caused by the fact that MCL is a less common lymphoma subtype compared to, for example, Hodgkin’s lymphoma with hence smaller patient populations and shorter follow-up periods to obtain a sufficiently large number of cases with the event of interest. However, 745 lesions were analyzed as 3D-volumetric whole lesion measurements and by utilizing advanced second-order radiomic features. A second limitation is the absence of test-retest studies to control confounding effects, like imaging parameters [61]. As a result, the findings of the paper are only explorative. Additional comparison with measurements in controls would be desirable and should be addressed in larger studies on this topic. Moreover, the variability in the timing of the subsequent imaging might have impacted the study results, as knowingly treatment-related changes underlie temporal influences. However, we believe that the findings are of great importance, as they strongly indicate the potential benefits of CT-based radiomics for MCL treatment. We plan to investigate this topic further and include the now ethically justified test-retest studies based on these findings. Thirdly, the manual definition of the tumour contour in the semi-automatic segmentation requires choosing the exact region of interest which requests proper expertise - a fully automatic segmentation would be much faster and more reliable. 

Given these promising preliminary findings, further work with larger datasets will analyze the strength of advanced radiomic parameters while adjusting for known predictors of outcomes to improve clinical outcomes by providing non-invasive, repeatable, and real-time surveillance to pave the way for personalized medicine, thus improving the standard of care for lymphoma. 

## 5. Conclusions

Our results suggest that CT-derived 3D radiomics has great potential as a complementary non-invasive biomarker for early prediction of relapse in MCL. In combination with the potential to confirm sustained remission, it might assist physicians in clinical management, especially if it is associated with an automatic classification tool. 

## Figures and Tables

**Figure 1 cancers-14-00393-f001:**
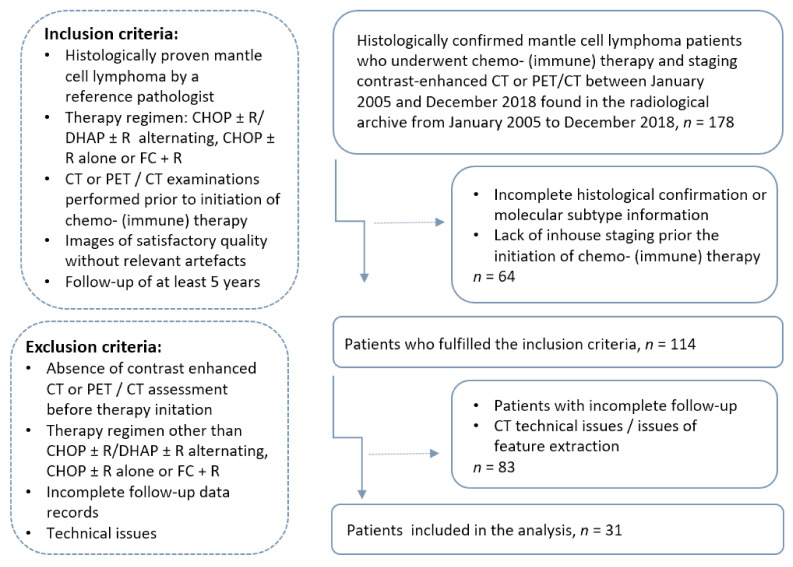
Recruitment pathway of the study.

**Figure 2 cancers-14-00393-f002:**
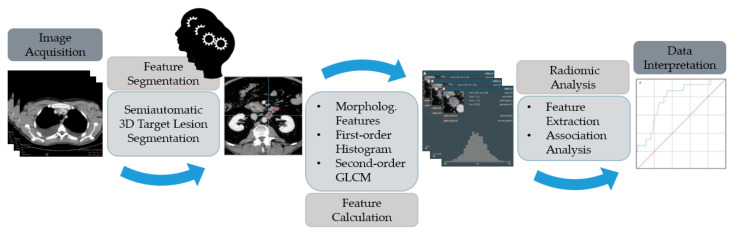
The schematic diagram for data processing and analysis. Size measurements, including 3D volume, were obtained as were first and second order textural features. Equations for these textural features can be accessed at https://pyradiomics.readthedocs.io/en/latest/features.html (accessed on 7 October 2021). Detailed settings of the extraction are listed in Appendix A, Table A1.

**Figure 3 cancers-14-00393-f003:**
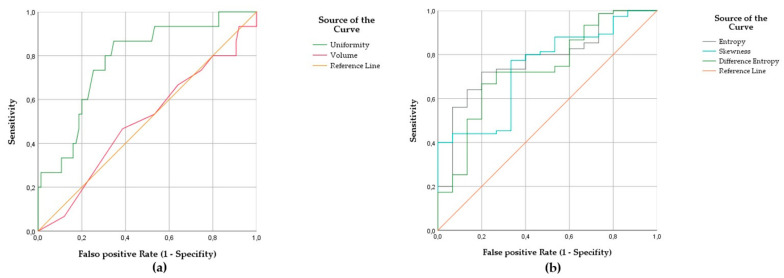
The receiver operating characteristics (ROC) curve of the feature “Uniformity” yielded an area under the curve (AUC) of 0.778, whereas “Volume” yielded an AUC of 0.500 for classification performance of CR and RD (**a**); ROC curves of the features “Entropy” (AUC 0.777), “Skewness” (AUC 0.738), and “Difference Entropy” (AUC 0.734) (**b**).

**Figure 4 cancers-14-00393-f004:**
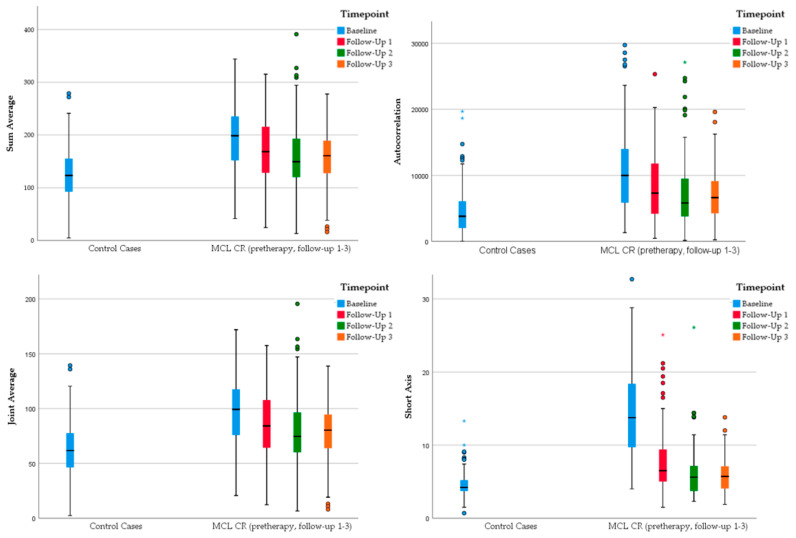
Grouped boxplots representing changes in specific radiomic parameters of MCL patients in complete remission (MCL CR) with no longer significant differences to the control cases with normal lymph nodes at the follow-up imaging at 6 and 12 months, respectively. * represents an outlier defined as a data point that is located outside 1.5 times the interquartile range above the upper quartile or below the lower quartile.

**Figure 5 cancers-14-00393-f005:**
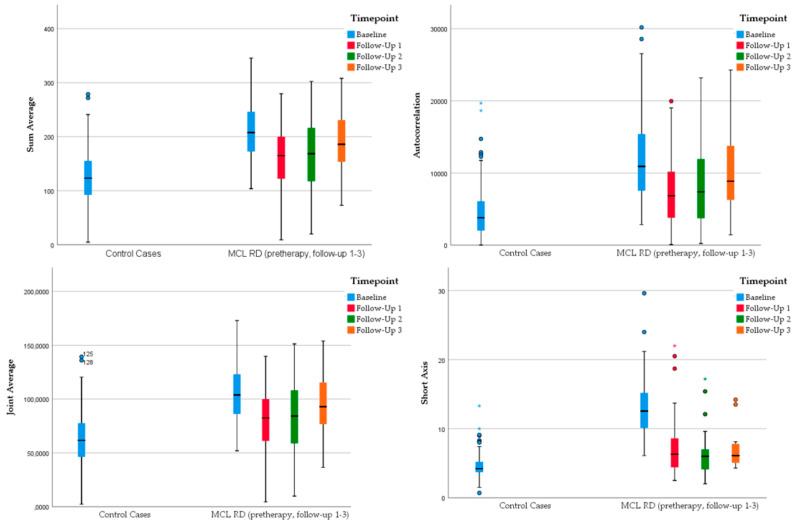
Grouped boxplots representing changes in specific radiomic parameters of MCL patients with relapse of disease (MCL RD) with continued significant differences to the control cases with normal lymph nodes at the pretherapy and all follow-up imaging with opposing values compared to the remission group at the follow-up imaging at 6 and 12 months, respectively. * represents an outlier defined as a data point that is located outside 1.5 times the interquartile range above the upper quartile or below the lower quartile.

**Table 1 cancers-14-00393-t001:** Baseline demographic, clinical, laboratory, and biological data of the entire MCL cohort and the reference group.

Characteristic	MCL Cohort (*n* = 31)	Reference Group (*n* = 39)
Age, median (range)	61.5 ± 9.7 years, (42–76 years)	64.9 ± 8.5 years, (47–81 years)
Male	26	21
Tissue Sample for histopathological Diagnosis	Lymph node: 12 patients Bone marrow: 11 patients Blood (liquid biopsy): 4 patients GI tract: 4 patients Nasal mucosa: 1 patient	PET/CT, clinical, and imaging follow-up over the next 5 years to confirm normal lymph node tissue and exclude hematological disease
Ann Arbor		not applicable
Stage I	0 patients	
Stage II	2 patients	
Stage III	5 patients	
Stage IV	25 patients	
MIPI (obtained/range)	7 patients (range 3–5.9)	not applicable
Radiotherapy	9 patients (29%)	not applicable
Therapy Regimen		
CHOP ± R alternatingwith DHAP ± R	19 patients (61.3%)	not applicable
CHOP ± R alone	10 patients (32.2%)	
FC + R	2 patients (6.5%)	

CHOP ± R (cyclophosphamide, doxorubicin, vincristine, and rituximab), DHAP± R (dexamethasone, high-dose cytarabine, cisplatin, and rituximab), R-FC (rituximab, fludarabine, and cyclophosphamide).

**Table 2 cancers-14-00393-t002:** Parameters evaluated for predictive performance, including morphologic measurements and radiomic features.

Radiomic Feature	AUC	Standard Error	*p*-Value	95% CI
Uniformity	0.788	0.063	0.001	0.655/0.902
Entropy	0.777	0.061	0.001	0.658/0.896
Skewness	0.738	0.066	0.004	0.608/0.867
Difference Entropy	0.734	0.071	0.004	0.595/0.874
SAD	0.620	0.072	0.145	0.479/0.760
LAD	0.532	0.083	0.695	0.369/0.695
Volume	0.500	0.084	1.000	0.226/0.664

AUC: area under the curve; CI: asymptotic confidence interval with lower and upper limits;. SAD: short axis; LAD: long axis.

**Table 3 cancers-14-00393-t003:** Sensitivity, specificity, accuracy, and cut-off values of the radiomic features best-suited for outcome prediction.

Radiomic Feature	Sensitivity	Specificity	Cut-Off for Relapse	Accuracy
Uniformity	87	65	≤0.0159	69
Entropy	80	72	≥6.2920	73
Skewness	67	77	≥−0.1890	76
Difference Entropy	80	67	≥5.2850	69

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
