# Peer review of "Longitudinal CT Imaging to Explore the Predictive Power of 3D Radiomic Tumour Heterogeneity in Precise Imaging of Mantle Cell Lymphoma (MCL)"

_cancers, 2022, doi:10.3390/cancers14020393_

Round 1

Reviewer 1 Report

Lisson et al describe the potential use of 3D radiomic tumor heterogeneity to determine prognosis in high risk mantle cell lymphoma patients. This was a retrospective analysis involving patients who diagnosed with high risk mantle cell lymphoma, treated with standard chemotherapy, who underwent the imaging that is being analyzed.  31 patients were identified out of 178 patients with mantle cell lymphoma. Negative prognostic radiographic factors identified were uniformity, entropy, skewness and difference entropy. None prognostic factors and positive prognostic factors were also identified, with the conclusion that further prospective studies were needed.

With regard to the individual sections:
Introduction: Well-written, no comments or additions.

Materials and methods: Within the scope of a retrospective analysis, images were reviewed by experienced radiologists in a blinded fashion as per the retrospective analysis. Statistical tools appear adequate for this type of study. No specific issues were noted in materials and methods.

Results: The results section is well-written and understandable. Tables and figures are adequately described and reflect the data presented in the narrative.

Discussion and conclusions: Understandable and well-written, appropriately identifying and discussing prognostic factors for progression and remission and proposing further prospective studies. 

Author Response

Dear Reviewer,

On behalf of my coauthors, I would like to thank you for taking the time to review and comment upon our manuscript entitled “Longitudinal CT Imaging to Explore the Prognostic Power of 3D Radiomic Tumor Heterogeneity in Precise Imaging of Mantle Cell Lymphoma” (ID: cancers-1503196).

Below we provide the point-by-point responses concerning the individual sections.

Thank you again for your thoughtful comments.

On behalf of all the co-authors

Sincerely,

Catharina Lisson

Clinician Scientist, Department of Diagnostic and Interventional Radiology,

University Hospital of Ulm, Albert-Einstein-Allee 23, 89081 Ulm, Germany

(+) 49 (0) 731 500 61171

[email protected]

Reviewer 2 Report

Lisson et al. reported their results on the application of radiomics in the context of prognostication of MCL. Overall, the data presented are new and fully original in particular considering the type of imaging proposed (i.e. CT scan) and complement those already published on radiomics in the context of positron emission tomography.

Furthermore, the paper is the first to the best of my knowledge to report the positive impact of the homogenous texture pattern of lymph nodes in MCL.

Main comments

  1. Line 45 “Despite the increasingly better understanding of the pathology and therapeutic options, the prognosis of MCL is generally poor, with 5-year survival rates of approximately 50%”, the sentence should be referenced. Moreover, it should be specified the population i.e transplant eligible or non-transplant eligible patients. I suppose the data refer to the transplant eligible population, but this should be clearly stated.
  2. Line 59-60 “This could be because unlike in chronic lymphocytic leukemia (CLL), MCL patients may 59 not have circulating clonal B cells or bone marrow involvement for MRD testing [8,13-15].”. This sentence must be clarified or modified. Liquid biopsy in CLL has been applied with important limitations due to the presence of circulation tumoral cells, which can impact the quality of ctDNA isolation. Moreover, as demonstrated by Moia et al (Br J Haematol. 2021 Oct;195(1):108-112.), this technique was not useful in the identification of tumour heterogeneity even in SLL, mainly due to the low quantity of circulating ct-DNA. All these data are in contrast with evidence from DLBCL and HL, where ct-DNA represent the most advanced prognostic factor available nowadays. As last point, even if the final paper has not yet been published, Lakhotia et al reported in 2018 the results of ct-DNA monitoring during DA_EPOCH-R chemotherapy in MCL, showing both a good correlation between the baseline ct-DNA level and the total tumour volume and the high ability of ct-DNA clearance after one cycle, to predict a better PFS in those clearing rapidly the ct-DNA (https://ashpublications.org/blood/article/132/Supplement%201/147/273048/Circulating-Tumor-DNA-Dynamics-during-Therapy)
  3. As last point of interest, I would like to better understand how much these data enhance the sensibility and correlate with the clinical prognostic stratification. In particular, it would be highly interesting to understand the correlation between this prognostic factor and MIPI and MIPI-c (not reported in the current paper, which integrates the clinical variables with the ki67 quantification).

Minor comments

  1. Line 53 “and leukocyte count.t”, there is a typo.
  2. Reference 14 is not on “liquid biopsy” but on the use of ddPCR in the longitudinal follow-up after chemotherapy for the detection of minimal residual disease to use a pre-emptive approach with Rituximab.

Author Response

Dear Reviewer,

On behalf of my coauthors, I would like to thank you for taking the time to review and comment upon our manuscript entitled “Longitudinal CT Imaging to Explore the Prognostic Power of 3D Radiomic Tumor Heterogeneity in Precise Imaging of Mantle Cell Lymphoma” (ID: cancers-1503196).

Below we provide the point-by-point responses.

Thank you again for your thoughtful comments.

On behalf of all the co-authors

Sincerely,

Catharina Lisson

Clinician Scientist, Department of Diagnostic and Interventional Radiology,

University Hospital of Ulm, Albert-Einstein-Allee 23, 89081 Ulm, Germany

(+) 49 (0) 731 500 61171

[email protected]

Reviewer 3 Report

Catharina Lisson and colleagues present a high quality and well-written experimental manuscript that describes longitudinal CT Imaging to explore the prognostic power of 3D radiomic tumor heterogeneity in precise imaging of mantle cell lymphoma.

Authors aimed to evaluate the prognostic performance of CT-derived 3D radiomics for MCL risk stratification. The secondary objective is to search for radiomic features associated with sustained remission. For that they investigated the potential of CT-derived 3D radiomics as a non-invasive prognostic biomarker to risk-stratify MCL patients, thus promoting precision imaging in clinical oncology.

Authors’ findings support the notion that CT-derived 3D radiomics improves prognostic estimation prognostication and might serve as a basis for a non-invasive biomarker to identify MCL patients at high risk of relapse; in combination with the potential to confirm sustained remission, it might assist physicians in the clinical management, especially if it is associated with an automatic classification tool.

Finally, authors conclude that this is the first study to investigate the potential value of longitudinal CT-derived 3D texture analysis for early prognostic estimation of MCL relapse based on radiomic changes during and after therapy.

Overall, the manuscript is highly valuable for the scientific community and should be accepted for publication.

======================

Other comments to authors:

1) Please check for typos throughout the manuscript.

2) Authors are kindly encouraged to cite the following article that describes various aspects of adoptive immunotherapies against lymphomas. DOI: 10.3390/cancers13040743 (PMID 33670139).

Author Response

Dear Reviewer,

On behalf of my coauthors, I would like to thank you for taking the time to review and comment upon our manuscript entitled “Longitudinal CT Imaging to Explore the Prognostic Power of 3D Radiomic Tumor Heterogeneity in Precise Imaging of Mantle Cell Lymphoma” (ID: cancers-1503196).

Below we provide the point-by-point responses.

Thank you again for your thoughtful comments.

On behalf of all the co-authors

Sincerely,

Catharina Lisson

Clinician scientist, Department of Diagnostic and Interventional Radiology,

University Hospital of Ulm, Albert-Einstein-Allee 23, 89081 Ulm, Germany

(+) 49 (0) 731 500 61171

[email protected]

Reviewer 4 Report

The authors used CT-derived 3D radiomics method to evaluate its value on the prognostic performance in mantle cell lymphomas. Though potential in the prognosis, but there are still some issues for the conclusion.

  1. The study is a retrospective observation which had some bias for the analysis. Such as not all the patients had the image follow up at 3, 6, and 12 months after therapy and the image follow up had a window of -/+ 4 weeks which might had bias on the conclusion.
  2. The number is small and more than half of fullfiled patients had been excluded from the study. 
  3. Lack of the validated group to confirmed the findings.

Author Response

Dear Reviewer,

On behalf of my coauthors, I would like to thank you for taking the time to review and comment upon our manuscript entitled “Longitudinal CT Imaging to Explore the Prognostic Power of 3D Radiomic Tumor Heterogeneity in Precise Imaging of Mantle Cell Lymphoma” (ID: cancers-1503196).

Below we provide the point-by-point responses.

On behalf of all the co-authors

Sincerely,

Catharina Lisson

Clinician scientist, Department of Diagnostic and Interventional Radiology,

University Hospital of Ulm, Albert-Einstein-Allee 23, 89081 Ulm, Germany

(+) 49 (0) 731 500 61171

[email protected]

Round 2

Reviewer 2 Report

Lisson et al. reported their results on the application of radiomics in the context of prognostication of MCL. As already expressed, the data presented are new and fully original in particular considering the type of imaging proposed (i.e. CT scan) and complement those already published on radiomics in the context of positron emission tomography. Furthermore, the paper is the first to the best of my knowledge to report the positive impact of the homogenous texture pattern of lymph nodes in MCL.

Moreover, in this revised version pf the paper, the Authors addressed all my comments and suggestions.